# Do Deep Neural Networks for Segmentation Understand Insideness?

## Abstract

Image segmentation aims at grouping pixels that belong to the same object or region. At the heart of image segmentation lies the problem of determining whether a pixel is inside or outside a region, which we denote as the "insideness" problem. Many Deep Neural Networks (DNNs) variants excel in segmentation benchmarks, but regarding insideness, they have not been well visualized or understood: What representations do DNNs use to address the long-range relationships of insideness? How do architectural choices affect the learning of these representations? In this paper, we take the reductionist approach by analyzing DNNs solving the insideness problem in isolation, *ie.* determining the inside of closed (Jordan) curves. We demonstrate analytically that state-of-the-art feed-forward and recurrent architectures can implement solutions of the insideness problem for any given curve. Yet, only recurrent networks could learn these general solutions when the training enforced a specific "routine" capable of breaking down the long-range relationships. Our results highlights the need for new training strategies that decompose the learning into appropriate stages, and that lead to the general class of solutions necessary for DNNs to understand insideness.

## 1 Introduction

Image segmentation is necessary for complete image understanding. A key component of image segmentation is to determine whether a pixel is inside or outside a region, *ie.* the "insideness" problem (Ullman, 1984; 1996). Deep Neural Networks (DNNs) have been tremendously successful in image segmentation benchmarks, but it is not well understood whether DNNs represent insideness or how.

Insideness has been overlooked in DNNs for segmentation since they have been mainly applied to the modality of "semantic segmentation", *ie.* labelling each pixel with its object category (Ronneberger et al., 2015; Yu & Koltun, 2016; Visin et al., 2016; Badrinarayanan et al., 2017; Chen et al., 2018b; Long et al., 2015; Lateef & Ruichek, 2019). In such cases, insideness is not necessary since a solution can rely only on object recognition. Yet, the recent need to solve more sophisticated visual tasks has fueled the development of DNNs with the ability to segment individual object instances, rather than object categories (Li et al., 2016; 2017; Song et al., 2018; Chen et al., 2018a; Hu et al., 2018; Maninis et al., 2018; Liu et al., 2018b; He et al., 2017). In these segmentation modalities, insideness plays a central role, especially when there are few cues besides the boundaries of the objects, *e.g.* when there is lack of texture and color, and objects are unfamiliar. Thus, insideness is necessary to achieve true generalization in image segmentation.

In this paper, we investigate derived and learned insideness-related representations in DNNs for segmentation. We take the reductionist approach by isolating insideness from other components in image segmentation. We analyze the segmentation of closed curves, similar to the methodology in Minsky & Papert's historic book *Perceptrons* (Minsky & Papert, 1969). In this way, we distill insideness to a minimum representation by eliminating other components.

We analytically demonstrate that two state-of-the-art network architectures, namely, DNNs with dilated convolutions (Yu & Koltun, 2016; Chen et al., 2018b) and convolutional LSTMs (ConvL-STMs) (Xingjian et al., 2015), among other networks, can exactly solve the insideness problem for any given curve with network sizes that are easily implemented in practice. The proofs draw on

algorithmic ideas from classical work on visual routines (Ullman, 1984; 1996), namely, the ray-intersection method and the coloring method, to derive equivalent neural networks that implement these algorithms. Then, in a series of experiments with synthetically generated closed curves, we evaluate the capabilities of these DNNs to learn the insideness problem. The experiments show that when using standard training strategies, the DNNs do not learn general solutions for insideness, even though these DNNs are sufficiently complex to capture the long-range relationships. The only network that achieves almost full generalization in all tested cases is a recurrent network with a training strategy designed to encourage a specific mechanism for dealing with long-range relationships.

These results add to the growing body of works that show that DNNs have problems in learning to solve some elemental visual tasks (Linsley et al., 2018; Liu et al., 2018a; Wu et al., 2018; Shalev-Shwartz et al., 2017). Shalev-Shwartz et al. (2017) introduced several tasks that DNNs can in theory solve, as it was shown mathematically, but the networks were unable to learn, not even for the given dataset, due to difficulties in the optimization with gradient descent. In contrast, the challenges we report for insideness are related to poor generalization rather than optimization, as our experiments show the networks succeed in solving insideness for the given dataset. Linsley et al. (2018) introduced new architectures that better capture the long-range dependencies in images. Here, we show that the training strategy has a big impact in capturing the long-range dependencies. Even if the DNNs we tested had the capacity to capture such long-range dependencies, they do not learn a general solution with the standard training strategies.

## 2 THE REDUCTIONIST APPROACH TO INSIDENESS

We now introduce the paradigm that will serve to analyze insideness-related representations in DNNs. Rather than using natural images, we use synthetic stimuli that solely contains a closed curve. In this way, we do not mix the insideness problem with other components of image segmentation found in natural images, *e.g.* self-similarity of segments at the level of object categories or parts, representation of the hierarchy of segments, *etc.*These components will be studied separately in future works, and finally put together to improve and understand how DNNs segment images.

Let $\boldsymbol{X} \in \{0,1\}^{N \times N}$ be an image or a matrix of size $N \times N$ pixels. We use $X_{i,j}$ and $(\boldsymbol{X})_{i,j}$, indistinguishably, to denote the value of the image in position $(i, j)$. We use this notation for indexing elements in any of the images and matrices that appear in the rest of the paper. Also, in the figures we use white and black to represent 0 and 1, respectively.

Insideness refers to finding which pixels are in the inside and which ones in the outside of a closed curve. We assume without loss of generality that there is only one closed curve in the image and that it is a digital version of a Jordan curve (Kong, 2001), *ie.* a closed curve without self-crosses nor self-touches and containing only horizontal and vertical turns, as shown in Fig. 1. We further assume that the curve does not contain the border of the image. The curve is the set of pixels equal to 1 and is denoted by $\mathcal{F}_{\boldsymbol{X}} = \{(i,j)|X_{i,j} = 1\}$.

The pixels in $\boldsymbol{X}$ that are not in $\mathcal{F}_{\boldsymbol{X}}$ can be classified into two categories: the inside and the outside of the curve (Kong, 2001). We define the segmentation of $\boldsymbol{X}$ as $\boldsymbol{S}(\boldsymbol{X}) \in \{0,1\}^{N \times N}$, where

$$S(\boldsymbol{X})_{i,j} = \begin{cases} 0 & \text{if } X_{i,j} \text{ is inside} \\ 1 & \text{if } X_{i,j} \text{ is outside} \end{cases}, \tag{1}$$

and for the pixels in $\mathcal{F}_{\boldsymbol{X}}$, the value of $S(\boldsymbol{X})_{i,j}$ can be either 0 or 1. Note that unlike object recognition, the definition of insideness is rigorously and uniquely determined by the input image itself.

The number of all digital Jordan curves is enormous even if the image size is relatively small, *e.g.* it is more than $10^{47}$ for the size $32 \times 32$ (App. A). In addition, insideness is a global problem; whether a pixel is inside or outside depends on the entire image, and not just on some local area around the pixel. Therefore, simple pattern matching, *ie.* memorization, is impossible in practice.

## 3 CAN DNNS FOR SEGMENTATION SOLVE INSIDENESS?

The universal approximation theorem (Cybenko, 1989) tells us that even a shallow neural network is able to solve the insideness problem. Yet, it could be that the amount of units is too large to

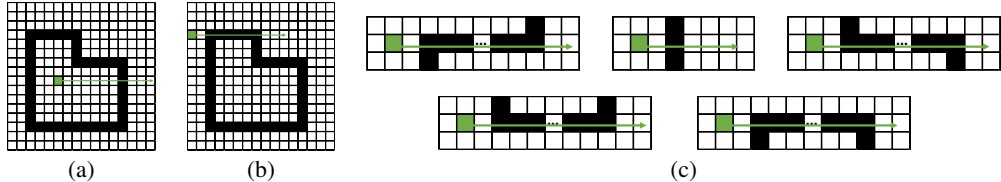

Figure 1: *Intersections of the Ray and the Curve.* (a) Example of ray going from one region to the opposite one when crossing the curve. (b) Example of ray staying in the same region after intersecting the curve. (c) All cases in which a ray could intersect a curve. In the three cases above the ray travels from one region to the opposite one, while in the two cases below the ray does not change regions.

be implementable in practice. In this Section, we introduce two DNN architectures that are able to solve the insideness problem at perfection and they are easily implementable in practice. One architecture is feed-forward with dilated convolutions (Yu & Koltun, 2016; Chen et al., 2018b) and the other is recurrent: a ConvLSTM (Xingjian et al., 2015).

### 3.1 FEED-FORWARD ARCHITECTURE WITH DILATED CONVOLUTIONS

Dilated convolutions facilitate capturing long-range dependencies which are key for segmentation (Yu & Koltun, 2016; Chen et al., 2018b). To demonstrate that there are architectures with dilated convolutions that can solve the insideness problem, we borrow insights from the ray-intersection method. The ray-intersection method (Ullman, 1984; 1996), also known as the crossings test or the even-odd test (Haines, 1994), is built on the following fact: Any ray that goes from a pixel to the border of the image alternates between inside and outside regions every time it crosses the curve. Therefore, the parity of the total number of such crossings determines the region to which the pixel belongs. If the parity is odd then the pixel is inside, otherwise it is outside (see Fig. 1a).

The definition of a crossing should take into account cases like the one depicted in Fig. 1b, in which the ray intersects the curve, but does not change region after the intersection. To address these cases, we enumerate all possible intersections of a ray and a curve, and analyze which cases should count as crossings and which ones should not. Without loss of generality, we consider only horizontal rays. As we can see in Fig. 1c, there are only five cases for how a horizontal ray can intersect the curve. The three cases at the top of Fig. 1c, are crosses because the ray goes from one region to the opposite one, while the two cases at the bottom (like in Fig. 1b) are not considered crosses because the ray remains in the same region.

Let $\vec{X}(i, j) \in \{0, 1\}^{1 \times N}$ be a horizontal ray starting from pixel $(i, j)$, which we define as

$$\vec{X}(i, j) = [X_{i,j}, X_{i,j+1}, X_{i,j+2}, \ldots, X_{i,N}, 0, \ldots, 0], \tag{2}$$

where zeros are padded to the vector if the ray goes outside the image, such that $\vec{X}(i, j)$ is always of dimension $N$. Let $\vec{X}(i, j) \cdot \vec{X}(i + 1, j)$ be the inner product of the ray starting from $(i, j)$ and the ray starting from the pixel below, $(i + 1, j)$. Note that the contribution to this inner product from the three cases at the top of Fig. 1c (the crossings) is odd, whereas the contribution from the other two intersections is even. Thus, the parity of $\vec{X}(i, j) \cdot \vec{X}(i + 1, j)$ is the same as the parity of the total number of crosses and determines the insideness of the pixel $(i, j)$, *ie.*

$$S(\boldsymbol{X})_{i,j} = \mathrm{parity}\left(\vec{X}(i, j) \cdot \vec{X}(i + 1, j)\right). \tag{3}$$

Dilated convolutions, also called atrous convolutions, are convolutions with upsampled kernels, which enlarge the receptive fields of the units but preserve the number of parameters of the kernel (Yu & Koltun, 2016; Chen et al., 2018b). In App. B we prove that equation 3 can be easily implemented with a neural network with dilated convolutions. The demonstration is based on implementing the dot product in equation 3 with multiple layers of dilated convolutions, as they facilitate capturing the information across the ray. The number of dilated convolutional layers is equal to the logarithm in base-2 of the image size, $N$. The dot product can also be implemented with two convolutional

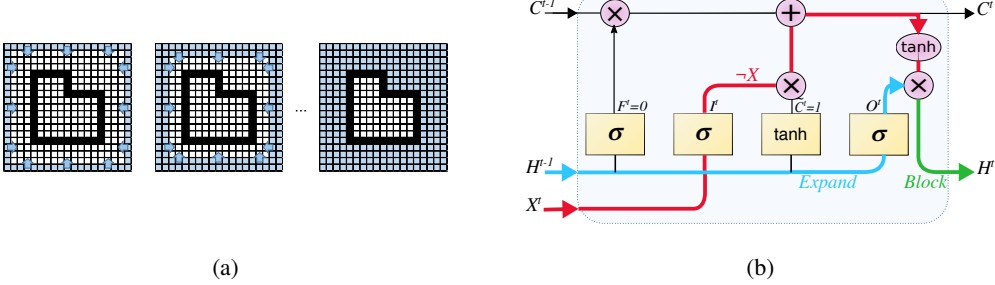

(a)                                                      (b)

Figure 2: *The Coloring Method with ConvLSTM*. (a) The coloring method consists of several iterations of the *coloring routine*, *ie.* expanding the outside region and blocking it on the curve. (b) Diagram of the ConvLSTM implementing the coloring method, we highlight the connections between layers that are used for insideness. $\neg \boldsymbol{X}$ denotes the element-wise "Boolean not" of $\boldsymbol{X}$.

layers, but with the drawback of using a long kernel of size $1 \times N$. The multiple dilated convolutions use kernels of size $3 \times 3$, and they are equivalent to the long kernel of $1 \times N$. Finally, the parity function in equation 3 is implemented by adapting the network introduced by Shalev-Shwartz et al. (2017), which yields a two layer convolutional network with $1 \times 1$ kernels.

Note that the proof introduces the smallest network we could find that solves the insideness problem with dilated convolutions. Larger networks than the one we introduced can also solve the insideness problem, as the network size can be reduced by setting kernels to zero and layers to implement the identity operation.

## 3.2 RECURRENT ARCHITECTURE: CONVOLUTIONAL LSTMS

Convolutional LSTM (ConvLSTM) (Xingjian et al., 2015) is another architecture designed to handle long-range dependencies. We now show that a ConvLSTM with just one kernel of size $3 \times 3$ is sufficient to solve the insideness problem. This is achieved by exploiting its internal back-projection of the LSTM, *ie.* the flow of information from a posterior layer to an anterior layer.

Our demonstration is inspired by the coloring method (Ullman, 1984; 1996), which is another algorithm for the insideness problem. This algorithm is based on the fact that neighboring pixels not separated by the curve are in the same region. We present a version of this method that will allow us to introduce the network with an LSTM. This method consists of multiple iterations of two steps: *(i)* expand the outside region from the borders of the image (which by assumption are in the outside region) and *(ii)* block the expansion when the curve is reached. The blocking operation prevents the outside region from expanding to the inside of the curve, yielding the solution of the insideness problem, as depicted in Fig. 2a. We call one iteration of expanding and blocking *coloring routine*.

We use $\boldsymbol{E}^t \in \{0,1\}^{N \times N}$ (expansion) and $\boldsymbol{B}^t \in \{0,1\}^{N \times N}$ (blocking) to represent the result of the two operations after iteration $t$. A The *coloring routine* can then be written as *(i)* $\boldsymbol{E}^t = \text{Expand}\left(\boldsymbol{B}^{t-1}\right)$ and *(ii)* $\boldsymbol{B}^t = \text{Block}\left(\boldsymbol{E}^t, \mathcal{F}_{\boldsymbol{X}}\right)$. Let $\boldsymbol{B}^{t-1}$ maintain a value of $1$ for all pixels that are known to be outside and $0$ for all pixels whose region is not yet determined or belong to the curve. Thus, we initialize $\boldsymbol{B}^0$ to have value $1$ (outside) for all border pixels of the image and $0$ for the rest. In step *(i)*, the outside region of $\boldsymbol{B}^{t-1}$ is expanded by setting also to $1$ (*outside*) its neighboring pixels, and the result is assigned to $\boldsymbol{E}^t$. Next, in step *(ii)*, the pixels in $\boldsymbol{E}^t$ that were labeled with a $1$ (*outside*) and belong to the curve, $\mathcal{F}_{\boldsymbol{X}}$, are reverted to $0$ (*inside*), and the result is assigned to $\boldsymbol{B}^t$. This algorithm ends when the outside region can not expand anymore, which is at most after $N^2$ iterations (worst case where each iteration expands the outside region by only one pixel). Therefore, we have $\boldsymbol{E}^{N^2} = \boldsymbol{S}(\boldsymbol{X})$.

In App. D we demonstrate that a ConvLSTM with one kernel applied on an image $\boldsymbol{X}$ can implement the coloring algorithm. In the following we provide a summary of the proof. Let $\boldsymbol{I}^t$, $\boldsymbol{F}^t$, $\boldsymbol{O}^t$, $\boldsymbol{C}^t$, and $\boldsymbol{H}^t \in \mathbb{R}^{N \times N}$ be the activations of the input, forget, and output gates, and cell and hidden states of a ConvLSTM at step $t$, respectively. By analyzing the equations of the ConvLSTM (equation 11 and equation 12 in App. D) we can see that the output layer, $\boldsymbol{O}^t$, back-projects to the hidden layer,

$H^t$. In the coloring algorithm, $E^t$ and $B^t$ are related in a similar manner. Thus, we define $O^t = E^t$ (expansion) and $H^t = \frac{1}{2}B^t$ (blocking). The $\frac{1}{2}$ factor is a technicality due to non-linearities, which is compensated in the output gate and has no relevance in this discussion.

We initialize $H^0 = \frac{1}{2}B^0$ (recall $B^0$ is 1 for all pixels in the border of the image and 0 for the rest). The output gate expands the hidden representations using one $3 \times 3$ kernel. To stop the outside region from expanding to the inside of the curve, $H^t$ takes the expansion output $O^t$ and sets the pixels at the curve's location to 0 (inside). This is the same as the element-wise product of $O^t$ and the "Boolean not" of $X$, which is denoted as $\neg X$. Thus, the blocking operation can be implemented as $H^t = \frac{1}{2}(O^t \odot \neg X)$, and can be achieved if $C^t$ is equal to $\neg X$. In Fig. 2b we depict these computations.

In App. D we show that the weights of a ConvLSTM with just one kernel of size $3 \times 3$ can be configured to reproduce these computations. A key component is that many of the weights use a value that tends to infinity. This value is denoted as $q$ and it is used to saturate the non-linearities of the ConvLSTM, which are hyperbolic tangents and sigmoids. Note that it is common in practice to have weights that asymptotically tend to infinity, *e.g.* when using the cross-entropy loss to train a network (Soudry et al., 2018). In practice, we found that saturating non-linear units using $q = 100$ is enough to solve the insideness problem for all curves in our datasets. Note that only one kernel is sufficient for ConvLSTM to solve the insideness problem, regardless of image size. Furthermore, networks with multiple stacked ConvLSTM and more than one kernel can implement the coloring method by setting unnecessary ConvLSTMs to implement the identity operation (App. D) and the unnecessary kernels to 0.

Finally, we point out that there are networks with a much lower complexity than LSTMs that can solve the insideness problem, although these networks rarely find applications in practice. In App. E, we show that a convolutional recurrent network as small as having one sigmoidal hidden unit per pixel, with a $3 \times 3$ kernel, can also solve the insideness problem for any given curve.

## 4    CAN DNNs FOR SEGMENTATION LEARN INSIDENESS?

After having identified DNNs that have sufficient complexity to solve the insideness problem, we focus on analyzing whether these solutions can be learnt from examples. We report experiments on synthetically generated Jordan curves. The goal of the network is to learn to predict for each pixel in the image whether it is inside or outside of the curve. In the following, we first describe the experimental setup, then analyze the generalization capabilities of the DNNs trained in standard manner and finally, analyse the advantages of the recurrent networks.

### 4.1    EXPERIMENTAL SETUP

**Datasets.** Given that the number of Jordan curves explodes exponentially with the image size, a procedure that could provide curves without introducing a bias for learning is unknown. We introduce three algorithms to generate different types of Jordan curves. For each dataset, we generate $95K$ images for training, $5K$ for validation and $10K$ for testing. All the datasets are constructed to fulfill the constraints introduced in Sec. 2. In addition, for testing and validation sets, we only use images that are dissimilar to all images from the training set. Two images are considered dissimilar if at least $25\%$ of the pixels of the curve are in different locations. In the following, we briefly introduce each dataset (see App. F for details). Fig. 3a, shows examples of curves for each dataset.
- *Polar Dataset* ($32 \times 32$ pixels): We use polar coordinates to generate this dataset. We randomly select the center of the figure and a random number of vertices that are connected with straight lines. The vertices are determined by their angles and distance with respect to the center of the figure. We generate 5 datasets with different maximum amount of vertices, namely, 4, 9, 14, 19 and 24, and refer to each dataset by this number, *e.g.* 24-Polar.
- *Spiral Dataset* ($42 \times 42$ pixels): The curves are generated by growing intervals of a spiral in random directions from a random starting point. The spiral has a random thickness at the different intervals.
- *Digs Dataset* ($42 \times 42$ pixels): We generate a rectangle of random size and then, we create "digs" of random thicknesses in the rectangle. The digs are created sequentially a random number of times.

**Evaluation metrics.** From the definition of the problem in Sec. 2, the pixels in the Jordan curve $\mathcal{F}_X$ are not evaluated. For the rest of the pixels, we use the following metrics:

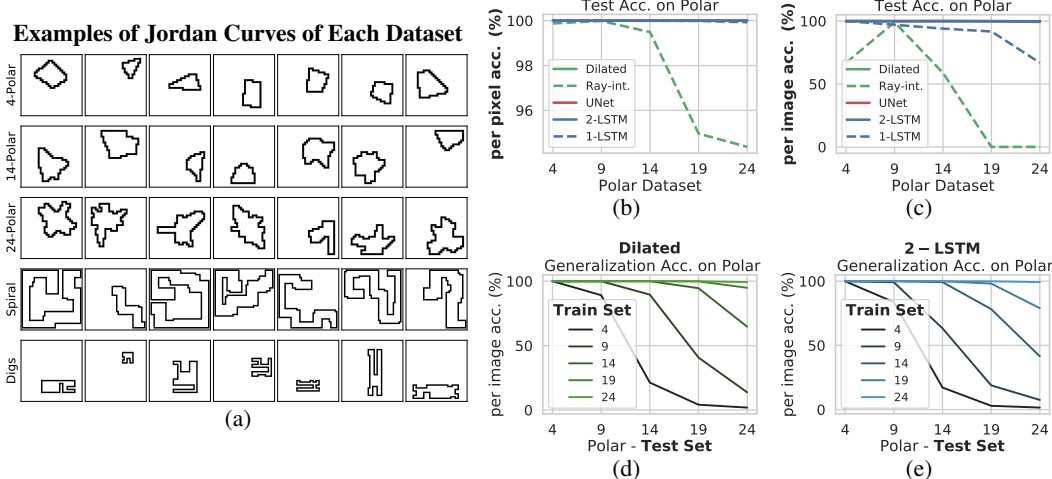

Figure 3: *Datasets and Results in Polar.* (a) Images of the curves used to train and test the DNNs. Each row correspond to a different dataset. Intra-dataset evaluation using (b) per pixel accuracy and (c) per image accuracy. Evaluation using the testing set of each Polar datasets for (d) *Dilated* and (e) *2-LSTM* networks.

- *Per pixel accuracy (%):* It is the average of the accuracy for inside and outside, evaluated separately. In this way, the metric weights the two categories equally, as there is an imbalance of inside and outside pixels.
- *Per image accuracy (%):* We use a second metric which is more stringent. Each image is considered correctly classified if all the pixels in the image are correctly classified.

**Architectures.** We evaluate the network architectures that we analyzed theoretically and also other relevant baselines:

- *Feed-forward Architectures:* We use the dilated convolutional DNN (*Dilated*) introduced in Sec. 3.1. We also evaluate two variants of *Dilated*, which are the Ray-intersection network (*Ray-int.*), which uses a receptive field of $1 \times N$ instead of the dilated convolutions, and a convolutional network (*CNN*), which has all the dilation factors set to $d = 1$. Finally, we also evaluate *UNet*, which is a popular architecture with skip connections and de-convolutions (Ronneberger et al., 2015).
- *Recurrent Architectures.* We test the ConvLSTM (*1-LSTM*) corresponding to the architecture introduced in Sec. 3.2. We initialize the hidden and cell states to $0$ (inside) everywhere except the border of the image which is initialized to $1$ (outside), such that the network can learn to color by expanding the outside region. We also evaluate a 2-layers ConvLSTM (*2-LSTM*) by stacking one *1-LSTM* after another, both with the initialization of the hidden and cell states of the *1-LSTM*. Finally, to evaluate the effect of such initialization, we test the *2-LSTM* without it (*2-LSTM w/o init.*), *ie.* with the hidden and cell states initialized all to $0$. We use backpropagation through time by unrolling $50$ time steps, for both training and testing.

**Learning.** The parameters are initialized using Xavier initialization (Glorot & Bengio, 2010). The derived parameters we obtained in the theoretical demonstrations obtain $100\%$ accuracy but we do not use them in this analysis as they are not learned from examples. The ground-truth consists on the insideness for each pixel in the image, as in equation 1. For all experiments, we use the cross-entropy with softmax as the loss function averaged accross pixels. Thus, the networks have two outputs per pixel (note that this does not affect the result that the networks are sufficiently complex to solve insideness, as the second output can be set to a constant threshold of $0.5$). We found that the cross-entropy loss leads to better accuracy than other losses. Moreover, we found that using a weighted loss improves the accuracy of the networks. The weight, which we denote as $\alpha$, multiplies the loss relative to inside, and $(1 - \alpha)$ multiplies the loss relative to outside. This $\alpha$ is a hyperparamter that we tune and can be equal to $0.1$, $0.2$ and $0.4$. We try batch sizes of $32$, $256$ and $2048$ when they fit in the GPUs' memory (12GB), and we try learning rates from $1$ to $10^{-5}$ (dividing by $10$). We train the networks for all the hyperparameters for at least $50$ epochs, and until there is no more improvement of the validation set loss. In the following, we report the testing accuracy for the hyperparameters that achieved the highest per image accuracy at the validation set. We test a large

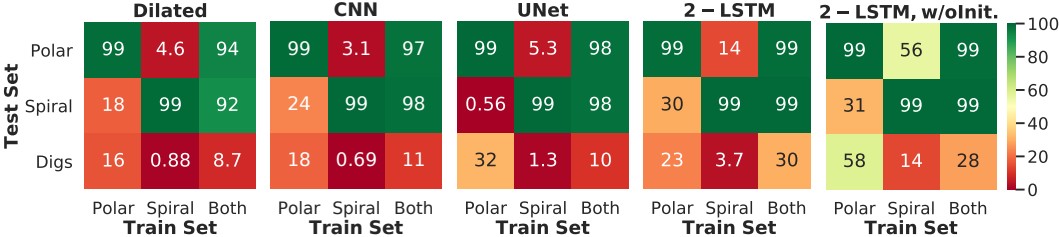

Figure 4: *Cross-dataset Results.* Evaluation of the networks trained in 24-Polar, Spiral and both 24-Polar and Spiral datasets. The tesing sets are 24-Polar, Spiral and Digs datasets.

set of hyperparameters (we trained several thousands of networks per dataset), which we report them in detail in App. G.

## 4.2 RESULTS

**Intra-dataset Evaluation.** In Fig.3b and c we show per pixel and per image accuracy for the networks trained on the same Polar dataset that are being tested. *Dilated*, *2-LSTM* and *UNet* achieve a testing accuracy very close to $100\%$, but *Ray-int.* and *1-LSTM* perform much worse. Training accuracy of *Ray-int.* and *1-LSTM* is the same as their testing accuracy (Fig. I.6a and b). This indicates an optimization problem similar to the cases reported by Shalev-Shwartz et al. (2017). Note that for the network with ConvLSTMs, we need two LSTMs to achieve an accuracy very close to $100\%$, even though one LSTM is sufficient to generalize, as we have previously shown. Similarly, both *Dilated* and *Ray-int.* are able to generalize, but only *Dilated* does so. It is an open question to understand why stochastic gradient descend performs so differently in each of these architectures can all generalize in theory. Finally, note that the per pixel accuracy is in most cases very high, and from now on, we only report the per image accuracy.

**Cross-dataset Evaluation.** We now evaluate if the networks that have achieved very high accuracies (*Dilated*, *2-LSTM* and *UNet*), have learnt the general solution of insideness that we introduced in Sec. 3. To do so, we train on one dataset and test on the different one. In Fig.3d and e, we observe that *Dilated* and *2-LSTM* do not generalize to Polar datasets with larger amount of vertices than the Polar dataset on which they were trained. Only if the networks are trained in 24-Polar, the networks generalize in all the Polar datasets. The same conclusions can be extracted for *UNet* (Fig. I.6c).

We further test generalization capabilities of these networks beyond the Polar dataset. In this more broad analysis, we also include the *CNN* and *2-LSTM w/o init*, by training them on 24-Polar, Spiral and both 24-Polar and Spiral, and test them on 24-Polar, Spiral and Digs separately. We can see in Fig. 4 that all tested networks generalize to new curves of the same family as the training set. Yet, the networks do not generalize to curves of other families. In Fig. I.12, we show qualitative examples of failed segmentations produced by networks trained on 24-Polar and Spiral and tested on the Digs dataset.

Furthermore, note that using a more varied training set ("Both") does not necessarily lead to better cross-dataset accuracy in all cases. For example, for *UNet* and *2-LSTM w/o init.*, training on Polar achieves better accuracy in Digs than when training on "Both". Also, for *Dilated*, training on "Both" harms its accuracy: the accuracy drops more than $6\%$ in 24-Polar and Spiral. In this case, the training accuracy is close to $100\%$, which indicates a problem of overfitting. We tried to address this problem by regularizing using weight decay, but it did not improve the accuracy (App. H).

**Visualization.** We now visualize the networks to study the representations learnt. In Fig. I.7, we analyze different units of *Dilated* trained on 24-Polar and Spiral. We display three units of the same kernel from the second and sixth layers, by showing the nine images in the testing set that produce the unit to be most active across all images (Zeiler & Fergus, 2014). For each image, we indicate the unit location by a gray dot. The visualizations suggest that units of the second layer are tuned to local features (*e.g.* Unit 19 is tuned to close parallel lines), while in layer 6 they are tuned to global ones (*e.g.* Unit 27 captures the space left in the center of a spiral). These features seem to capture characteristics of the curves in the training set. This is quite different from the representations that

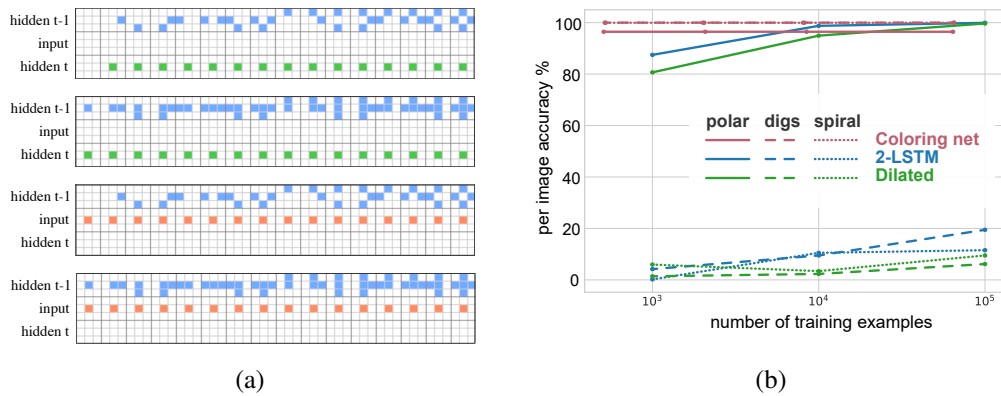

(a)                                                                                          (b)

Figure 5: *Learning the Coloring Routine*. (a) 64 possible inputs and outputs of the training set of the *Coloring Net* for the relevant inputs. The *Coloring Net* is trained to reproduce one step of the *Coloring Routine*. (b) Per image accuracy for different datasets training with different amounts of examples. *2-LSTM* and *Dilation* are trained on 24-Polar.

we derived theoretically, which accumulate the number of crossings in a ray from each pixel. This is further supported by visualizing the feature maps in Fig. I.9.

In Fig. I.11, we display the feature maps of *2-LSTM* trained on 24-Polar and Spiral. The figure shows the feature maps of the layers at different time steps. We can see that the network expands the borders of the image, which have been initialized to outside. Yet, it also expands the curve, which is not what our analytical solution does (Fig. I.10). This explains why this representation does not generalize to new datasets, because it is not possible to know the direction where to expand the curve without having *a priori* knowledge of the curve.

### 4.3 LEARNING THE COLORING ROUTINE IN ISOLATION

We now analyse a property of the coloring method that is relevant for learning: the *coloring routine* does not contain long-range relationships because it just takes into account $3 \times 3$ neighbourhoods. The long-range relationships are captured by applying the *coloring routine* multiple times. The standard training strategy enforces the ground-truth after the last step, and hence, requires learning the full long-range relationships at once. Yet, if we decompose the learning of insideness into learning the *coloring routine* in isolation, the problem becomes much simpler as it only requires learning an operation in a $3 \times 3$ neighbourhood.

The *coloring routine* can be learned by enforcing to each step the ground-truth produced by the routine, rather than waiting until the last step. The inputs of a step are the image and the hidden state of the previous step. Recall that the *coloring routine* determines that a pixel is outside if there is at least one neighbor assigned to outside that is not at the curve border. All input cases (64) are depicted in Fig. 5a, leaving the irrelevant inputs for the *coloring routine* at 0. During learning, such irrelevant pixels are assigned randomly a value of 0 or 1.

We have done an architecture search to learn the *coloring routine*. We could not make any of the previously introduced *LSTM* networks fit a step of the *coloring routine* due to optimization problems. Yet, we found a simple network that succeeded: a convolutional recurrent neural network with a sigmoidal hidden layer and an output layer that backprojects to the hidden layer. The kernel sizes are $3 \times 3$ and $1 \times 1$ for the hidden and output layers, respectively, and we use 5 kernels. We call this network *Coloring Net*. Observe that this network is sufficiently complex to solve the insideness problem, because it is the network introduced in App. E with an additional layer and connections.

The *Coloring Net* reaches 0 training error about 40% of the times after randomly initializing the parameters. After training the *Coloring Net* in one step, we unroll it and apply it to images of Jordan curves. In Fig. 5b we report the accuracy of the *Coloring Net* in the 24-*Polar*, *Spiral* and *Digs* datasets, for different amounts of training examples (generated through adding more variations of the irrelevant inputs). We compare the results with the *2-LSTM* and *Dilation* networks previously

introduced, trained on 24-*Polar*. We can see that with less than $1000$ examples the *Coloring Net* is able to generalize to any of the datasets, while the other networks do not. This demonstrates the great potential of decomposing the learning to facilitate the emergence of the routine.

## 5   Conclusions and Future Work

We have shown that DNNs with dilated convolutions and convolutional LSTM that are implementable in practice are sufficiently complex to solve the insideness problem for any given curve. When using the standard training strategies, the units in these networks become specialized to detect characteristics of the curves in the training set and only generalize to curves of the same family as the training, even when using large number of training examples. Yet, we found that when simple recurrent networks are supervised to learn the *coloring routine*, which does not contain long-range relationships, the general solution for the insideness problem emerged using orders of magnitude less data.

This raises the question of whether these findings can be translated to improvements of segmentation methods for natural images. The following experiment suggests that state-of-the-art methods for image segmentation suffer from learning general solutions to the insideness problem. We evaluate two off-the-shelf methods, namely DEXTR (Maninis et al., 2018) for instance segmentation and DeepLabv3+ (Chen et al., 2018c) for semantic segmentation, which have been trained on PASCAL VOC 2012 (Everingham et al.) and ImageNet (Russakovsky et al., 2015). These methods fail to determine the insideness for a vast majority of curves, even after fine-tuning in the *Both* dataset (Deeplabv3+ achieved $36.58\%$ per image accuracy in *Both* dataset and $2.18\%$ in *Digs*, see implementation details and qualitative examples in App. J). Thus, extending these methods with recurrent connections and a training strategy that could capture the coloring routine, could help increase their segmentation accuracy, especially for different conditions on which they have been trained.

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

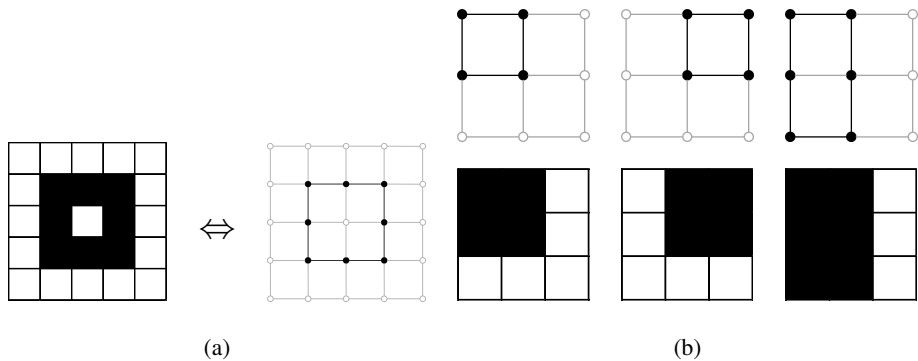

Figure A.1: *Subgraph Representations of Figures.* (a) A figure in an image of size $5 \times 5$ pixels (left) and its subgraph representation in a grid graph of $5 \times 5$ vertices (right). (b) Cycles that are not digital Jordan curves (top) and their correspondents (bottom).

## A  NUMBER OF DIGITAL JORDAN CURVES

We now introduce a procedure to derive a lower bound of the number of Jordan curves in an image. We represent an image of size $N \times N$ pixels by using a grid graph (square lattice) with $N \times N$ vertices. We employ 4-adjacency for black pixels and corresponding grid points, and 8-adjacency for white pixels and their counterpart. Then, a curve (the set of black pixels) corresponds to a subgraph of the base grid graph (Fig. A.1a).

In this representation, a digital Jordan curve is defined as a subgraph specified by a sequence of vertices $(v_0, v_1, \ldots, v_L)$ satisfying the following conditions (Rosenfeld, 1970; Kong, 2001):

1. $L \geq 4$,

2. $v_r = v_s$ if and only if $r = s$, and

3. $v_r$ is 4-adjacent to $v_s$ if and only if $r \equiv s \pm 1 \pmod{L + 1}$.

Note that conditions 1 and 2 defines a cycle (Harary, 1969) in a grid graph. Therefore, any digital Jordan curve is a cycle but not vice versa. Figure A.1b shows examples of cycles that are not digital Jordan curves.

The numbers of all cycles in grid graphs of different sizes were computed up to $27 \times 27$ vertices (Iwashita et al., 2013; A14), and we utilize this result to get lower bounds for the number of digital Jordan curves with the following considerations.

Although a cycle in a grid graph is not necessarily a digital Jordan curve as shown above, we can obtain a digital Jordan curve in a larger image from any cycle by "upsampling" as shown in Fig. A.2a. Note that there are other digital Jordan curves than the ones obtained in this manner (therefore we get a lower bound with this technique). See Fig. A.2b for examples.

We also consider "padding" shown in Fig. A.3 to assure that a digital Jordan curve does not contain the border of a image (this is what we assume in the main body of the paper).

Taking everything into consideration, we can obtain a lower bound of the number of digital Jordan curves in an $N \times N$ image that does not contain border pixels utilizing the above-mentioned result (Iwashita et al., 2013; A14), upsampling and padding. Table 1 shows lower bounds obtained in this way. For example, starting with the row 2 of (Karavaev & Iwashita) in (A14) (this represents the number of all cycles in the grid graph with $3 \times 3$ vertices), we get a lower bound 13 for the number of digital Jordan curves in $5 \times 5$ images by considering the upsampling and get the same number as a lower bound for the number of digital Jordan curves that do not contain border pixels in $7 \times 7$ images by considering the padding.

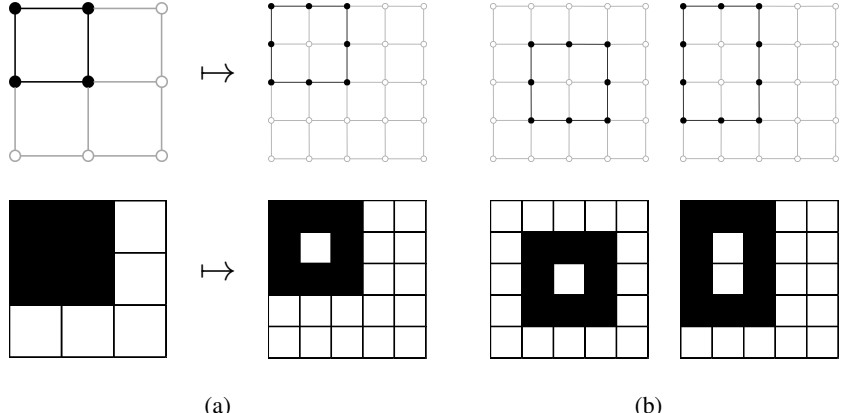

(a)                                    (b)

Figure A.2: *"Upsampling" Operation and Its Limitations.* (a) Depiction of "upsampling" operation. (b) Digital Jordan curves that cannot be obtained by the upsampling shown in (a). (Left) The issue is the place of the digital Jordan curve. We can get the same curve on the upper-left, upper-right, lower-left and lower-right corners but cannot get the one in the center. (Right) The issue is the length of the side. We cannot get a side with 4 vertices (4 pixels) nor with even number vertices (pixels) in general.

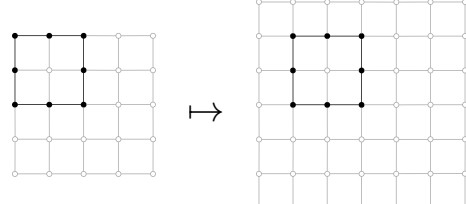

Figure A.3: *Depiction of "Padding".*

# B INSIDENESS WITH DILATED CONVOLUTIONAL NETWORKS

We first introduce a feed-forward convolutional DNN for which there exist parameters that reproduce equation 3. Then, we show that one of the layers in this network can be better expressed with multiple dilated convolutions.

## B.1 CONVOLUTIONAL DNN TO IMPLEMENT THE RAY-INTERSECTION METHOD

The smallest CNN that we found that implements the ray-intersection method has 4-layers. As we show in the following, the first two layers compute $\vec{X}(i,j) \cdot \vec{X}(i+1,j)$, and the last two layers compute the parity. We use $\boldsymbol{H}^{(k)} \in \mathbb{R}^{N \times N}$ and $[\cdot]_+$ to denote the activations of the units at the $k$-th layer, and the *ReLU* activation function, respectively. Fig. B.4a depicts the architecutre.

**First and Second Layer: Inner product.** For the sake of simplicity, we only use horizontal rays, but the network that we introduce can be easily adapted to any ray direction. The first layer implements all products needed for the inner products across all rays in the image, *ie.* $X_{i,j} \cdot X_{i+1,j}, \forall(i,j)$. Note that there is exactly one product per pixel, and each product can be reused for multiple rays. For convenience, $H_{i,j}^{(1)}$ represents the product in pixel $(i,j)$, *ie.* $H_{i,j}^{(1)} = X_{i,j} \cdot X_{i+1,j}$. Since the input consists of binary images, each product can be reformulated as

$$H_{i,j}^{(1)} = \begin{cases} 1 & \text{if} \quad X_{i,j} = X_{i+1,j} = 1 \\ 0 & \text{otherwise} \end{cases}. \tag{4}$$

This equality can be implemented with a *ReLU*: $H_{i,j}^{(1)} = [1 \cdot X_{i,j} + 1 \cdot X_{i+1,j} - 1]_+$. Thus, $\boldsymbol{H}^{(1)}$ is a convolutional layer with a $2 \times 1$ kernel that detects the intersections shown in Fig. 1c. This

Table 1: *Lower bounds (LBs) of the number of digital Jordan curves in $N \times N$ images that do not contain border pixels.*

| $N$ | 5 | 7 | 9 | $\cdots$ | 31 | 33 | 35 | $\cdots$ | 55 |
|---|---|---|---|---|---|---|---|---|---|
| LB | 1 | 13 | 213 | $\cdots$ | $1.203 \times 10^{47}$ | $1.157 \times 10^{54}$ | $3.395 \times 10^{61}$ | $\cdots$ | $6.71 \times 10^{162}$ |

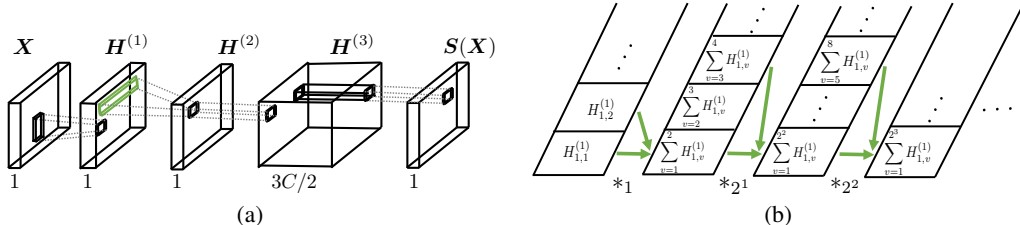

(a)                                                                    (b)

Figure B.4: *The Ray-Intersection Network.* (a) The receptive field colored in green has size $1 \times N$, and it can be substituted by an equivalent network composed of multiple dilated convolutions. (b) The $1 \times N$ kernel of the ray-intersection network is equivalent to multiple dilated convolutional layers. The figure shows an horizontal ray of the activations of several layers, starting from the first layer $H^{(1)}$. The green arrows indicate the locations in the ray that lead to the desired sum of the activations, *ie.* the sum of the ray.

layer can also be implemented with a standard convolutional layer with a $3 \times 3$ kernel, by setting the unnecessary elements of the kernel to $0$.

The second layer sums over the products of each ray. To do so, we use a kernel of dimension $1 \times N$ with weights equal to $1$ and bias equal to $0$, *ie.* $H_{i,j}^{(2)} = \boldsymbol{1}_{1 \times N} \cdot H_{i,j}^{(1)} = \vec{X}(i,j) \cdot \vec{X}(i+1,j)$, in which $\boldsymbol{1}_{I \times J}$ denotes the matrix of size $I \times J$ with all entries equal to $1$. Zero-padding is used to keep the kernel size constant across the image.

Note that the shape of the kernel, $1 \times N$, is not common in the DNN literature. Here, it is necessary to capture the long-range dependencies of insideness. We show in the next subsection that it can be substituted by multiple layers of dilated convolutions.

**Third and Fourth Layers: Parity.** To calculate the parity of each unit's value in $\boldsymbol{H}^{(2)}$, we borrow the DNN introduced by Shalev-Shwartz et al. (2017) (namely, Lemma 3 in the supplemental material of the paper). This network obtains the parity of any integer bounded by a constant $C$. The network has $3C/2$ hidden *ReLUs* and one output unit, which is $1$ if the input is even, $0$ otherwise (see App. C for details).

We apply this parity network to all units in $\boldsymbol{H}^{(2)}$ via convolutions, reproducing the network for each unit. Since a ray through a closed curve in an $N \times N$ image can not have more than $N$ crossings, $C$ is upper bounded by $N$. Thus, the third layer has $3N/2$ kernels, and both the third and output layer are convolutions with a $1 \times 1$ kernel. At this point we have shown that the DNN explained above is feasible in practice, as the number of kernels is $O(N)$, and it requires no more than $4$ convolutional layers with *ReLUs*. The network has a layer with a kernel of size $1 \times N$, and next we show that this layer is equivalent to several layers of dilated convolutions of kernel size $3 \times 3$.

### B.2 DILATED CONVOLUTIONS TO IMPLEMENT THE $1 \times N$ KERNEL

We use $*_d$ to denote a dilated convolution, in which $d$ is the dilation factor. Let $\boldsymbol{H} \in \mathbb{R}^{N \times N}$ be the units of a layer and let $\boldsymbol{K} \in \mathbb{R}^{k \times k}$ be a kernel of size $k \times k$. A dilated convolution is defined as follows: $(\boldsymbol{H} *_d \boldsymbol{K})_{i,j} = \sum_{-\lfloor k/2 \rfloor \leq v,w \leq \lfloor k/2 \rfloor} H_{i+dv,j+dw} \cdot K_{v,w}$, in which $H_{i+dv,j+dw}$ is $0$ if $i + dv$ or $j + dw$ are smaller than $0$ or larger than $N$, *ie.* we abuse notation for the zero-padding. Note that in the dilated convolution the kernel is applied in a sparse manner, every $d$ units, rather than in consecutive units. See Yu & Koltun (2016); Chen et al. (2018b) for more details on dilated convolutions.

Recall the kernel of size $1 \times N$ is set to $\mathbf{1}_{1 \times N}$ so as to perform the sum of the units corresponding to the ray in the first layer, *ie.* $\sum_{0 \leq v < N} H^{(1)}_{i,j+v}$. We can obtain this long-range sum with a series of dilated convolutions using the following $3 \times 3$ kernel:

$$\boldsymbol{K} = \begin{bmatrix} 0 & 0 & 0 \\ 0 & 1 & 1 \\ 0 & 0 & 0 \end{bmatrix}. \tag{5}$$

First, we apply this $\boldsymbol{K}$ to the $\boldsymbol{H}^{(1)}$ through $*_1$ in order to accumulate the first two entries in the ray, which yields: $\left(\boldsymbol{H}^{(2)}\right)_{i,j} = \left(\boldsymbol{H}^{(1)} *_1 \boldsymbol{K}\right)_{i,j} = \sum_{0 \leq v \leq 1} H^{(1)}_{i,j+v}$. As shown in Fig. B.4b, to accumulate the next entries of the ray, we can apply $\boldsymbol{K}$ with a dilated convolution of dilation factor $d = 2$, which leads to $\left(\boldsymbol{H}^{(3)}\right)_{i,j} = \sum_{0 \leq v < 4} H^{(1)}_{i,j+v}$. To further accumulate more entries of the ray, we need larger dilation factors. It can be seen in Fig. B.4b that these dilation factors are powers of 2, which yield the following expression:

$$\left(\boldsymbol{H}^{(l)}\right)_{i,j} = \left(\boldsymbol{H}^{(l-1)} *_{2^{l-2}} \boldsymbol{K}\right)_{i,j} = \sum_{0 \leq v < 2^{l-2}} H^{(1)}_{i,j+v}. \tag{6}$$

Observe that when we reach layer $l = \log_2(N) + 2$, the units accumulate the entire ray of length $N$, *ie.* $\sum_{0 \leq v < N} H^{(1)}_{i,j+v}$. Networks with dilation factors $d = 2^l$ are common in practice, *e.g.* Yu & Koltun (2016) uses these exact dilation factors.

In summary, DNNs with dilated convolutions can solve the insideness problem and are implementable in practice, since the number of layers and the number of kernels grow logarithmically and linearly with the image size, respectively.

## C  PARITY NETWORK BY SHALEV-SHWARTZ ET AL. (2017)

To calculate the parity of each unit's value in $\boldsymbol{H}^{(2)}$, we borrow the DNN introduced by Shalev-Shwartz *et al.* (namely, Lemma 3 in the supplemental material of Shalev-Shwartz et al. (2017)). This network obtains the parity of any integer bounded by a constant $C$. The network has $\frac{3C}{2}$ hidden units with *ReLUs* and one output unit, which is 1 if the input is even, 0 otherwise. Since such a network requires an upper bound on the number whose parity is being found, we define $C$ as the maximum number of times that a horizontal ray can cross $\mathcal{F}_{\boldsymbol{X}}$. This number can be regarded as an index to express the complexity of the shape.

There is a subtle difference between the network introduced by Shalev-Shwartz et al. (2017) and the network we use in the paper. In Shalev-Shwartz et al. (2017), the input of the network is a string of bits, but in our case, the sum is done in the previous layer, through the dilated convolutions. Thus, we use the network in Shalev-Shwartz et al. (2017) after the sum of bits is done, *ie.* after the first dot product in the first layer in Shalev-Shwartz et al. (2017).

To calculate the parity, for each even number between 0 and $C$ (0 included), $\{2i \mid 0 \leq i \leq \lfloor C/2 \rfloor\}$, the network has three hidden units that threshold at $(2i - \frac{1}{2})$, $2i$ and $(2i + \frac{1}{2})$, *ie.* $-0.5, 0, 0.5, 1.5, 2, 2.5, 3.5, 4, 4.5, \ldots$ The output layer linearly combines all the hidden units and weights each triplet of units by 2, $-4$ and 2. Observe that when the input is an odd number, the three units in the triplet are either all below or all above the threshold. The triplets that are all below the threshold contribute 0 to the output because the units are inactive, and the triplets that are all above the threshold also contribute 0 because the linear combination is $2(2i - \frac{1}{2}) - 4(2i) + 2(2i + \frac{1}{2}) = 0$. For even numbers, the triplet corresponding to that even number has one unit below, equal and above the threshold. The unit that is above the threshold contributes 1 to the output, yielding the parity function.

## D  COLORING ROUTINE WITH A CONVOLUTIONAL LSTM

Here we prove that a ConvLSTM can implement the coloring routine, namely, the iteration of the expansion and the blocking operations. A ConvLSTM applied on an image $\boldsymbol{X}$ is defined as the fol-

lowing set of layers (see Xingjian et al. (2015) for a comprehensive introduction to the ConvLSTM):

$$\boldsymbol{I}^t = \sigma \left( \boldsymbol{W}^{xi} * \boldsymbol{X} + \boldsymbol{W}^{hi} * \boldsymbol{H}^{t-1} + \boldsymbol{b}^i \right), \tag{7}$$

$$\boldsymbol{F}^t = \sigma \left( \boldsymbol{W}^{xf} * \boldsymbol{X} + \boldsymbol{W}^{hf} * \boldsymbol{H}^{t-1} + \boldsymbol{b}^f \right), \tag{8}$$

$$\tilde{\boldsymbol{C}}^t = \tanh \left( \boldsymbol{W}^{xc} * \boldsymbol{X} + \boldsymbol{W}^{hc} * \boldsymbol{H}^{t-1} + \boldsymbol{b}^c \right), \tag{9}$$

$$\boldsymbol{C}^t = \boldsymbol{F}^t \odot \boldsymbol{C}^{t-1} + \boldsymbol{I}^t \odot \tilde{\boldsymbol{C}}^t, \tag{10}$$

$$\boldsymbol{O}^t = \sigma \left( \boldsymbol{W}^{xo} * \boldsymbol{X} + \boldsymbol{W}^{ho} * \boldsymbol{H}^{t-1} + \boldsymbol{b}^o \right), \tag{11}$$

$$\boldsymbol{H}^t = \boldsymbol{O}^t \odot \tanh \left( \boldsymbol{C}^t \right), \tag{12}$$

where $\boldsymbol{I}^t$, $\boldsymbol{F}^t$, $\boldsymbol{C}^t$, $\boldsymbol{O}^t$ and $\boldsymbol{H}^t \in \mathbb{R}^{N \times N}$ are the activation of the units of the input, forget, cell state, output and hidden layers at $t$, respectively. Note that $\boldsymbol{C}^t$ has been decomposed with the help of the auxiliary equation defining $\tilde{\boldsymbol{C}}^t$. Note also that each of these layers use a different set of weights that are applied to $\boldsymbol{X}$ and to $\boldsymbol{H}^t$ denoted as $\boldsymbol{W} \in \mathbb{R}^{N \times N}$ with superindices that indicate the connections between layers, *e.g.* $\boldsymbol{W}^{xi}$ are the weights that connect $\boldsymbol{X}$ to $\boldsymbol{I}$. Similarly, the biases are denoted as $\boldsymbol{b} \in \mathbb{R}^{N \times N}$ with the superindices indicating the layers. The symbols $*$ and $\odot$ denote the (usual, not dilated) convolution and the element-wise product, respectively. Finally, $\sigma$ and $\tanh$ are the sigmoid and the hyperbolic tangent, which are used as non-linearities.

We can see by analyzing equation 11 and equation 12 that the output layer, $\boldsymbol{O}^t$, back-projects to the hidden layer, $\boldsymbol{H}^t$. In the coloring algorithm, $\boldsymbol{E}^t$ and $\boldsymbol{B}^t$ are related in a similar manner. Thus, we define $\boldsymbol{O}^t = \boldsymbol{E}^t$ (expansion) and $\boldsymbol{H}^t = \frac{1}{2} \boldsymbol{B}^t$ (blocking), as depicted in Fig. 2b. The $\frac{1}{2}$ factor will become clear below, and it does not affect the correctness. We initialize $\boldsymbol{H}^0 = \frac{1}{2} \boldsymbol{B}^0$ (recall $\boldsymbol{B}^0$ is 1 for all pixels in the border of the image and 0 for the rest). We now show how to implement the iteration of the expansion and the blocking operations with the ConvLSTM:

**(i) Expansion, $\boldsymbol{O}^t$:** We set the output layer in equation 11 in the following way:

$$\boldsymbol{O}^t = \sigma \left( 2q \boldsymbol{1}_{3 \times 3} * \boldsymbol{H}^{t-1} - \frac{q}{2} \boldsymbol{1}_{N \times N} \right). \tag{13}$$

Note that this layer does not use the input, and sets the convolutional layer $\boldsymbol{W}^{ho}$ to use a $3 \times 3$ kernel that is equal to $2q \boldsymbol{1}_{3 \times 3}$, in which $q$ is a scalar constant, and the bias equal to $-\frac{q}{2} \boldsymbol{1}_{N \times N}$. For very large values of $q$, this layer expands the outside region. This can be seen by noticing that for a unit in $\boldsymbol{H}^{t-1}$, if at least one neighbor has value $1/2$, then $O^t_{i,j} = \lim_{q \to \infty} \sigma(q) = 1$. Also, when all neighbouring elements of the unit are 0, then no expansion occurs because $O^t_{i,j} = \lim_{q \to \infty} \sigma(-\frac{q}{2}) = 0$.

**(ii) Blocking, $\boldsymbol{H}^t$:** To stop the outside region from expanding to the inside of the curve, $\boldsymbol{H}^t$ takes the expansion output $\boldsymbol{O}^t$ and sets the pixels at the curve's location to 0 (inside). This is the same as the element-wise product between $\boldsymbol{O}^t$ and the element-wise "Boolean not" of $\boldsymbol{X}$, which is denoted as $\neg \boldsymbol{X}$. Thus, the blocking operation can be implemented as $\boldsymbol{H}^t = \frac{1}{2} (\boldsymbol{O}^t \odot \neg \boldsymbol{X})$. Observe that if $\boldsymbol{C}^t = \neg \boldsymbol{X}$, this is equal to equation 12 of the LSTM, because $\tanh(0) = 0$ and $\tanh(1) = 1/2$, *ie.*

$$\boldsymbol{H}^t = \boldsymbol{O}^t \odot \tanh \left( \boldsymbol{C}^t \right) = \frac{1}{2} \boldsymbol{O}^t \odot \neg \boldsymbol{X}. \tag{14}$$

We can obtain $\boldsymbol{C}^t = \neg \boldsymbol{X}$, by imposing $\boldsymbol{I}^t = \neg \boldsymbol{X}$ and $\boldsymbol{C}^t = \boldsymbol{I}^t$, as shown in Fig. 2b. To do so, let $\boldsymbol{W}^{xi} = -q \boldsymbol{1}_{1 \times 1}$, $\boldsymbol{W}^{hi} = \boldsymbol{0}_{N \times N}$, and $\boldsymbol{b}^i = \frac{q}{2} \boldsymbol{1}_{N \times N}$, and equation 8 becomes the following expression:

$$\boldsymbol{I}^t = \lim_{q \to \infty} \sigma \left( -q \boldsymbol{1}_{1 \times 1} * \boldsymbol{X} + \frac{q}{2} \boldsymbol{1}_{N \times N} \right). \tag{15}$$

Observe that when $q$ tends to infinity, we have $I^t_{i,j} = \lim_{q \to \infty} \sigma(\frac{q}{2}) = 1$ when $X_{i,j} = 0$ and $I^t_{i,j} = \lim_{q \to \infty} \sigma(-\frac{q}{2}) = 0$ when $X_{i,j} = 1$, which means $\boldsymbol{I}^t = \neg \boldsymbol{X}$. Next, to obtain $\boldsymbol{C}^t = \boldsymbol{I}^t$, we set $\boldsymbol{W}^{xf} = \boldsymbol{W}^{hf} = \boldsymbol{W}^{xc} = \boldsymbol{W}^{hc} = \boldsymbol{0}_{N \times N}$, $\boldsymbol{b}^f = -q \boldsymbol{1}_{N \times N}$ and $\boldsymbol{b}^c = q \boldsymbol{1}_{N \times N}$. This leads to the desired result:

$$\boldsymbol{F}^t = \lim_{q \to \infty} \sigma \left( -q \boldsymbol{1}_{N \times N} \right) = \boldsymbol{0}_{N \times N}, \tag{16}$$

$$\tilde{\boldsymbol{C}}^t = \lim_{q \to \infty} \tanh \left( q \boldsymbol{1}_{N \times N} \right) = \boldsymbol{1}_{N \times N},$$

$$\boldsymbol{C}^t = \boldsymbol{0}_{N \times N} \odot \boldsymbol{C}^{t-1} + \boldsymbol{I}^t \odot \boldsymbol{1}_{N \times N} = \boldsymbol{I}^t = \neg \boldsymbol{X}. \tag{17}$$

Thus, the coloring method can be implemented with a network as small as one ConvLSTM with one kernel. A network with more than one kernel and multiple stacked ConvLSTM can also solve the insideness problem for any given curve. The kernels that are not needed to implement the coloring method can be just set to $0$ and the ConvLSTM that are not needed should implement the identity operation, *ie.* the output layer is equal to the input. To implement the identity operator, equation 11 can be rewritten in the following way:

$$\boldsymbol{O}^t = \lim_{q \to \infty} \sigma \left( q \boldsymbol{1}_{1 \times 1} * \boldsymbol{X} - \frac{q}{2} \boldsymbol{1}_{N \times N} \right) \tag{18}$$

where $\boldsymbol{W}^{ho} = \boldsymbol{0}_{1 \times 1}$ is to remove the connections with the hidden units, and $q$ is the constant that tends to infinity. Observe that if $X_{i,j} = 1$, then $\boldsymbol{O}^t = \lim_{q \to \infty} \sigma(q/2) = 1$. If $X_{i,j} = 0$, then $\boldsymbol{O}^t = \lim_{q \to \infty} \sigma(-q/2) = 0$. Thus, the ConvLSTM implements the identity operation.

## E    COLORING ROUTINE WITH A SIGMOIDAL CONVOLUTIONAL RNN

There are other recurrent networks simpler than a ConvLSTM that can also implement the coloring algorithm. We introduce here a convolutional recurrent network that uses sigmoids as non-linearities. Since it is a convolutional network, for the sake of simplicity we just describe the operations done to obtain an output pixel in a step. The network has only one hidden layer, which also corresponds to the output of the network. Let $\{h_k^t\}_{k \in \mathcal{N}_{i,j}}$ be the hidden state of the output pixel indexed by $i, j$ and its 4-neighbourhood, at step $t$. Let $X_{i,j}$ be the only relevant input image pixel. A necessary condition is that the outputs of the sigmoid should asymptotically be close to $0$ or $1$, otherwise the coloring routine would fade after many steps. It is easy to check that $h_{i,j}^{t+1} = \sigma \left( q \left( \sum_{k \in \mathcal{N}_{ij}} h_k^t - 5 X_{i,j} - 1/2 \right) \right)$ implements the coloring routine, where $q$ is the factor that ensures saturation of the sigmoid.

## F    DATASET GENERATION

In Fig. F.5, we show more examples of curves in the datasets. In the following we provide a more detailed description of the algorithms to generate the curves:

- *Polar Dataset* ($32 \times 32$ pixels): We use polar coordinates to generate this dataset. We randomly select the center of the figure and a random number of vertices that are connected with straight lines. These lines are constrained to follow the definition of digital Jordan curve in Sec. 2 in the main paper (and App. A in this supplementary material). The vertices are determined by their angles, which are randomly generated. The distance with respect to the center of the figure are also randomly generated to be between 3 to 14 pixels away from the center.

We generate 5 datasets with different maximum amount of vertices, namely, 4, 9, 14, 19 and 24. We refer to each of these datasets as *Polar* with a prefix with the amount of vertices.

- *Spiral Dataset* ($42 \times 42$ pixels): The curves in these data set are generated from a random walk. First, a starting position is chosen uniformly at random from $[10, 20] \times [10, 20]$. Then, a segment of the spiral is built in the following way: a random direction (up, down, left, right) and a random length $r$ from 3 to 10 are chosen so that the walk is extended by turning $r$ pixels in the given direction. However, such extension can only happen if adding a random thickness $t \in \{1, 2, 3, 4\}$ to both sides of this segment does not cause self intersections. These segments are added repeatedly until there is no space to add a new segment without violating the definition of a Jordan curve.

- *Digs Dataset* ($42 \times 42$ pixels): We generate a rectangle of random size and then, we create "digs" of random thicknesses in the rectangle. The number of "digs" is a random number between 1 to 10. The digs are created sequentially and they are of random depth (between 1 pixel to the length of the rectangle minus 2 pixels). For each new "dig", we made sure to not cross previous digs by adjusting the depth of the "dig".

## G    HYPERPARAMETERS

In this Section we report all the tried hyperparameters for all architectures. In all cases, the convolutional layers use zero-padding.

**Examples of Jordan Curves of Each Dataset**

Figure F.5: *Datasets.* Images of the curves used to train and test the DNNs. Each row correspond to a different dataset.

*- Dilated Convolution DNN (Dilated):* This network was introduced in Sec. 3.1. We use the same hyperparameters as in Yu & Koltun (2016): $3 \times 3$ kernels, a number of kernels equal to $2^l \times \{2, 4, 8\}$, where $l$ is the number of layers and ranges between $8$ to $11$, with $d = 2^l$ (the first layer and the last two layers $d = 1$). The number of kernels in the layer that calculates the parity can be $\{5, 10, 20, 40, 80\}$.

*- Ray-intersection network (Ray-int.):* This is the architecture introduced in Sec. 3.1, which uses a receptive field of $1 \times N$ instead of the dilated convolutions. The rest of the hyperparameters are as in *Dilated*.

*- Convolutional DNN (CNN):* To analyze the usefulness of the dilated convolutions, we use the *Dilated* architecture with all dilation factors $d = 1$. Also, we try adding more layers than in *Dilated*, up to $25$.

*- UNet:* This is a popular architecture with skip connections and de-convolutions. We use similar hyperparameters as in Ronneberger et al. (2015): starting with $64$ kernels ($3 \times 3$) at the first layer and doubling this number after each max-pooling; a total of $1$ to $3$ max-pooling layers in all the network, that are placed after sequences of $1$ or $2$ convolutional layers.

*- Convolutional LSTM (1-LSTM):* This is the architecture with just one ConvLSTM, introduced in Sec. 3.2. The number of time steps is fixed to $50$. We initialize the hidden and cell states to $0$ (inside) everywhere except the border of the image which is initialized to $1$ (outside).

*- 2-layers Convolutional LSTM (2-LSTM):* We stack one convolutional LSTM after another. The first LSTM has $64$ kernels, and the hidden and cell states are initialized as in the 1-LSTM.

*- 2-layers Convolutional LSTM without initialization (2-LSTM w/o init.):* this is the same as the *2-LSTM* architecture the hidden and cell states are initialized to $0$ (outside).

# H ADDITIONAL EXPERIMENTS OF FEED-FORWARD NETWORKS

In Fig. 4, we have observed that *Dilated* trained on both *24-Polar* and *Spiral* datasets, obtains a test accuracy of less than $95\%$ on these datasets while the accuracy in the training set is very close to $100\%$. We added weight decay in all the layers in order to regularize the network. We tried values between $10^{-5}$ to $1$, scaling by a factor of $10$. In all these experiments we have observed overfitting except for a weight decay of $1$, in which the training never converged.

Also, note that the *CNN* does not have this overfitting problem. Yet, the number of layers needed is $25$, which is more than the double than for *Dilated*, which is $9$ layers. We added more layers to *Dilated* but the accuracy did not improve.

# I ADDITIONAL FIGURES AND VISUALIZATIONS

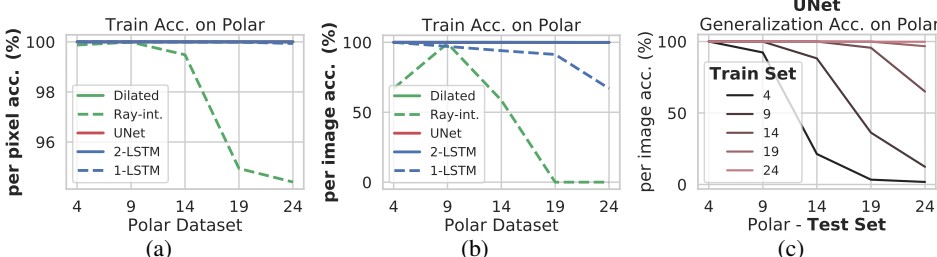

(a)  (b)  (c)

Figure I.6: *Training Accuracy in the Polar Dataset.* Intra-dataset evaluation using (a) per pixel accuracy and (b) per image accuracy on the training set, which are very similar to the test accuracy reported in Fig. 3b and c. (c) Intra-dataset evaluation of *Unet*.

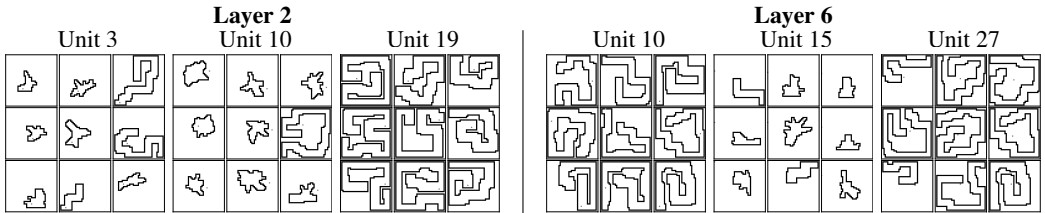

Figure I.7: *Visualization of the Units Learnt by* Dilation. Each block are the 9 images that produce the maximum activation of a units in a convolutional across the test set. The gray dot indicates the location of the unit. Fig. I.8 shows more examples.

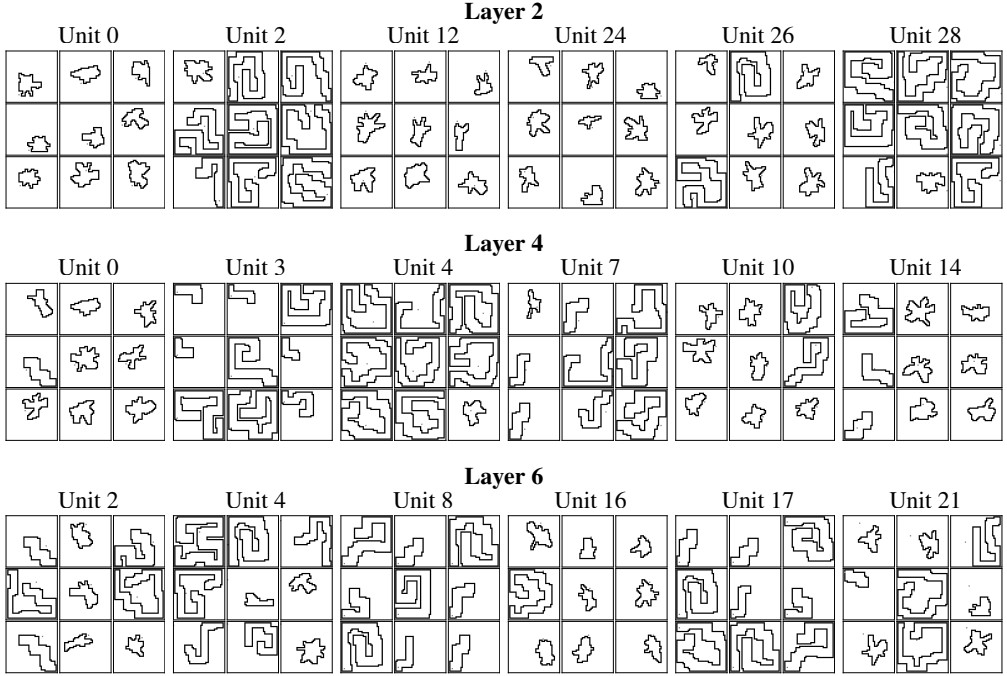

Figure I.8: *More examples of Visualization of the Units Learnt by* Dilation.

**Examples of Feature Maps of the Dilated Conv. DNN**

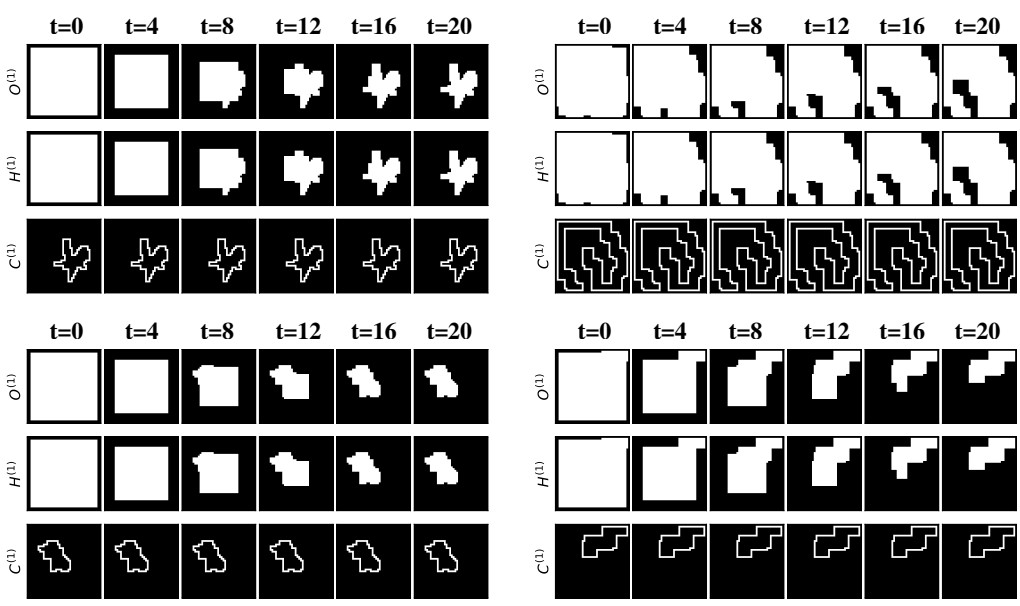

Figure I.9: *Visualization of the Feature Maps of the DNN with Dilated Convolutions.* We display several feature maps at different layers (each row in each block is a different layer). We can see that the first layers detect lower level features such as edges with specific orientations, while the later layers capture long-range dependencies of the curve relative to insideness.

Figure I.10: *Visualization of Convolutional LSTM with the Mathematically Derived Parameters.* We can see that only the border of the image (outside) is propagated, and not the curve, as in the learnt solution.

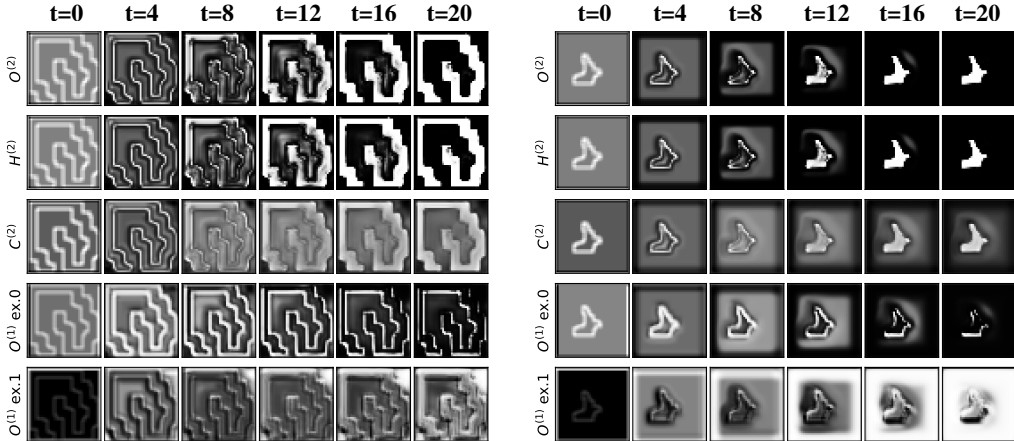

Figure I.11: *Activation Maps of the Learnt Representations by* 2-LSTM. Each row corresponds to a different layer and each colum to a different time step.

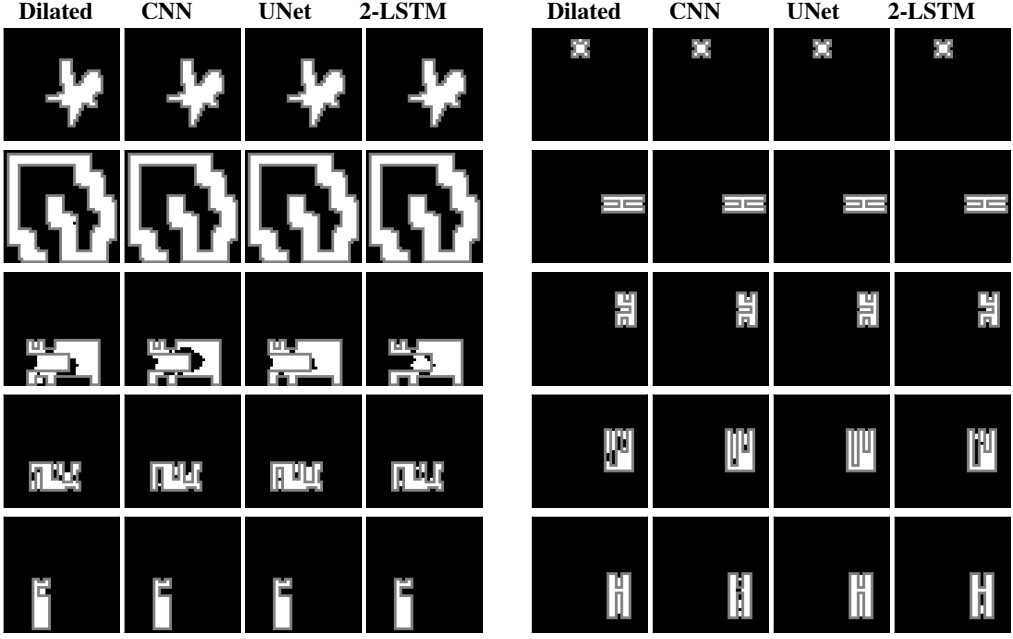

Figure I.12: *Qualitative Examples from the Digs dataset.* Networks trained in 24-Polar and Spiral dataset fail to segment in the Digs dataset.

## J    NETWORKS PRE-TRAINED ON NATURAL IMAGES

We chose two state-of-the-art networks on Instance Segmentation, DEXTR (Maninis et al., 2018) and DeepLabv3+ (Chen et al., 2018c), to investigate their ability in solving the insideness problem.

**DEXTR.** Deep Exteme Cut (DEXTR) is a neural network used for interactive instance segmentation. We use the pre-trained model on PASCAL VOC 2012 (Everingham et al.) and show some of the qualitative results in Fig. J.13.

**DeepLabv3+.** This architecture extends DeepLabv3 (Chen et al., 2018b) by utilizing it as an encoder network and adding a decoder network to refine segmentation boundaries. The encoder employs dilated convolution and Atrous Spatial Pyramid Pooling module for feature extraction. we use DeepLabv3+ with Xception backbone pretrained on PASCAL VOC 2012, and fine-tune its last layer with Polar and Spiral datasets for training. The ratio of input image spatial resolution to encoder output image is referred to as output stride and varies according to dilation rates. We use output strides of 8 and 16 as suggested in the paper; loss weight ($\alpha$) of 0.1, 0.2 and 0.4; and initial learning rates from 0.1 to $10^{-5}$ (dividing by 10). We train the network on Polar and Spiral datasets until there is no improvement of the accuracy at the validations set, and we then reduce the learning rate by a ratio of 10 and stop at the next plateau of the validation set accuracy.

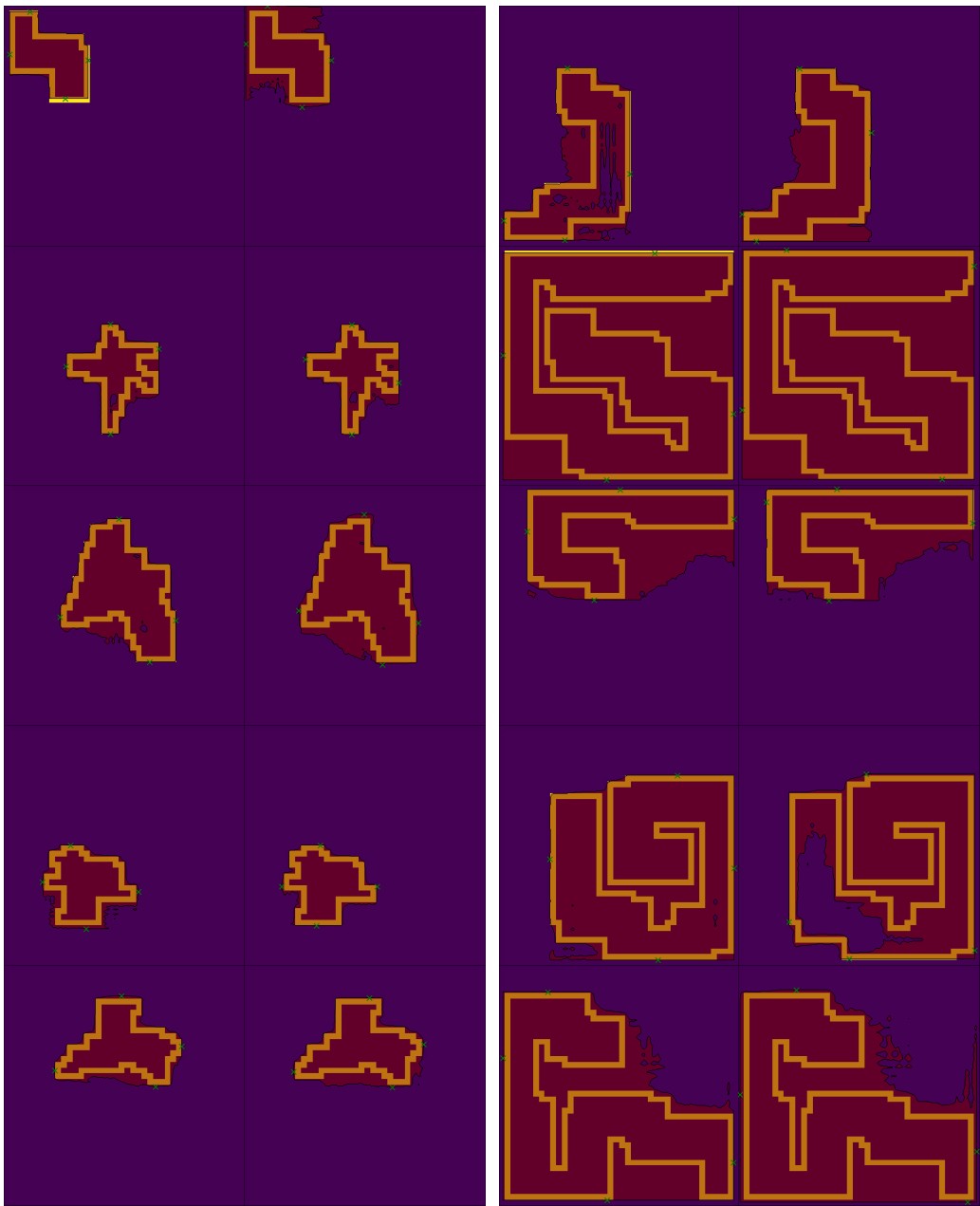

Figure J.13: *Qualitative Results with DEXTR on the Polar Dataset.* We use the publicly available pre-trained DEXTR model (Maninis et al., 2018). DEXTR uses 4 points marked by the user (indicated with crosses). We report the best found points, two examples of them per image.

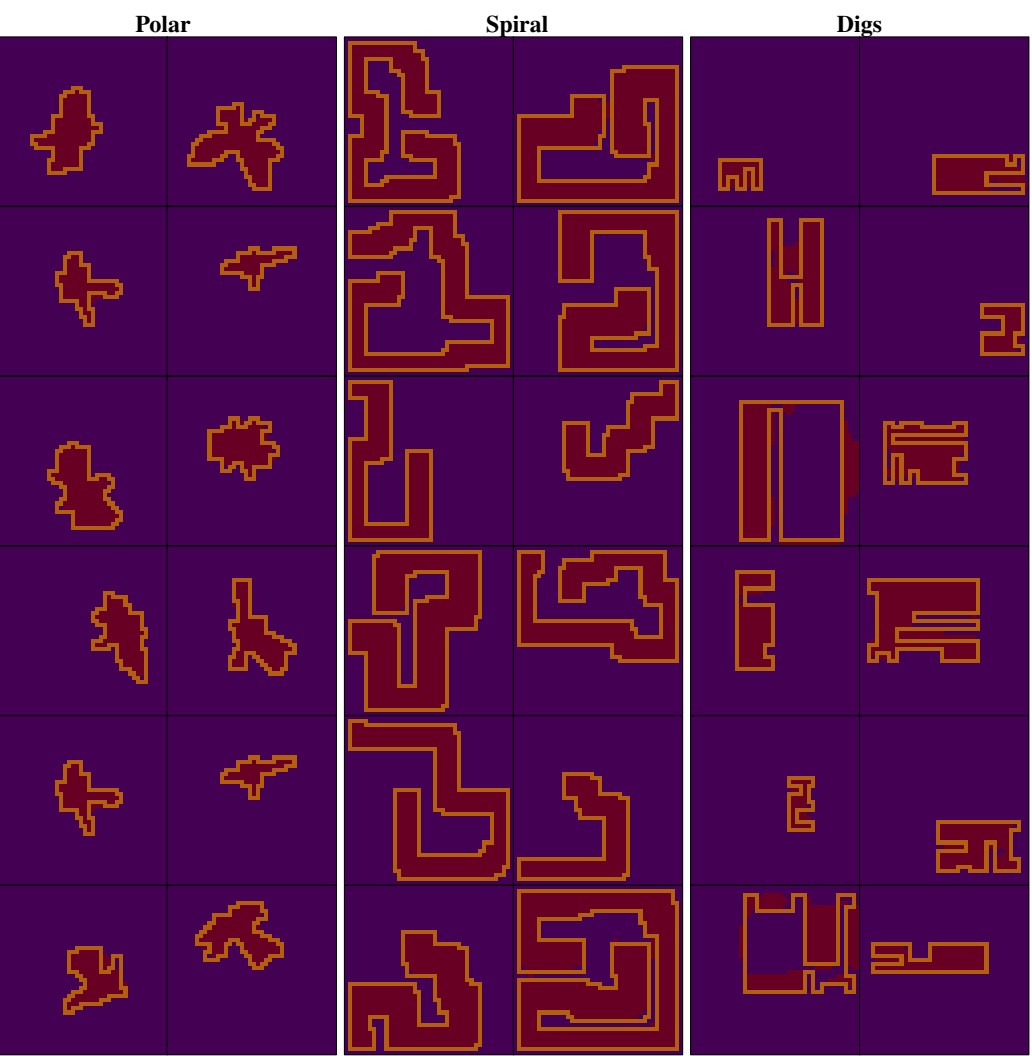

Figure J.14: *Results of DeepLabv3+ on Polar, Spiral and Digs Datasets.* The network is fine-tuned on Polar and Spiral. The results show that the network predicts well most of the pixels except in the borders. For the cross-dataset evaluations in the Digs dataset, the network is not able to generalize.

