# OpenReview forum: "Do Deep Neural Networks for Segmentation Understand Insideness?"
_ICLR.cc/2020/Conference — Reject_

### Official Review · AnonReviewer2 · 2019-10-21
**Official Blind Review #2**

**Rating:** 6

**Review:**

This submission introduces a new concept, termed insideness, to study semantic segmentation in deep learning era. The authors raise many interesting questions, such as (1) Does deep neural networks (DNN) understand insideness? (2) What representations do DNNs use to address the long-range relationships of insideness? (3) How do architectural choices affect the learning of these representations? This work adopts two popular networks, dilated DNN and ConvLSTM, to implement solutions for insideness problem in isolation. The results can help future research in semantic segmentation for the models to generalize better.

I give an initial rating of weak accept because I think (1) This paper is well written and well motivated. (2) The idea is novel, and the proposed "insideness" seems like a valid metric. This work is not like other segmentation publications that just propose a network and start training, but perform some deep analysis about the generalization capability of the existing network architectures. (3) The experiments are solid and thorough. Datasets are built appropriately for demonstration purposes. All the implementation details and results can be found in appendix. (4) The results are interesting and useful. It help other researchers to rethink the boundary problem by using the insideness concept. I think this work will have an impact in semantic segmentation field.

I have one concern though. The authors mention that people will raise the question of whether these findings can be translated to improvements of segmentation methods for natural images. However, their experiments do not answer this question. Fine-tuning DEXTR and Deeplabv3+ on the synthetic datasets can only show the models' weakness, but can't show your findings will help generalize the model to natural images. Adding an experiment on widely adopted benchmark datasets, such as Cityscapes, VOC or ADE20K, will make the submission much stronger.







**Experience Assessment:**

I have published one or two papers in this area.

**Review Assessment: Checking Correctness Of Derivations And Theory:**

I carefully checked the derivations and theory.

**Review Assessment: Checking Correctness Of Experiments:**

I carefully checked the experiments.

**Review Assessment: Thoroughness In Paper Reading:**

I read the paper thoroughly.

---

> ### Author Response · Authors · 2019-11-12
> **Rebuttal**
>
> We appreciate the many positive aspects that R#2 highlighted about the paper. It is very encouraging. Thank you.
>
> Regarding the only concern, we agree with this reviewer that the experiment with off-the-shelf models is confusing as it is placed in the future work section and does not guarantee that our findings can improve segmentation in natural images. To avoid this confusion, we have moved the experiment to the experiments section, where is useful to emphasize the lack of generalization of existing DNNs for segmentation. We have also added in the future work section that our findings leave several open questions that require future research, such as learning insideness to improve segmentation in natural images, in cartoons and sketches, and in other contexts, as well as improving other tasks that require spatial understanding.

---

### Official Review · AnonReviewer1 · 2019-11-01
**Official Blind Review #1**

**Rating:** 6

**Review:**

The paper shows that deep-nets can actually learn to solve the problem of "what is inside a curve" by using a sort of progressive filling of the space outside the curve. The paper suceeds in explaining that and in pointing out the limitations if standard learning to address this problem.

However,
(1) the proposed demonstration is based in ideal (continuous, noiseless) curves. What would happen in actual (discontinuous, noisy) curves?. What implications does this has in the requirements of the network.
(2) I think more connections to classical and current algorithms are required. For instance:
(2.1) The proposed demonstration (and the arguments of Ullman) reminds me of classical watersheed algorithms [see 2.1]. What is the gain of using deep networks with regard to rather old techniques?. Advantages are not clear in the text.
(2.2) What about connections to recent algorithms of automatic fill-in of images of contours based on conditional GANs [see 2.2]. It seems that these recent techniques already solved the "insideness" problem and even learnt how to fill the inside in sensible ways...
Then, what is the gain of the proposed approach?.

REFERENCES:

[2.1] Fundamenta Informaticae 41 (2001) 187–228
The Watershed Transform:  Definitions, Algorithms and Parallelization Strategies
Jos B.T.M. Roerdink and Arnold Meijster
http://www.cs.rug.nl/~roe/publications/parwshed.pdf

[2.2] Unpaired Image-to-Image Translation using Cycle-Consistent Adversarial Networks
Jun-Yan Zhu, Taesung Park, Phillip Isola, Alexei A. Efros
ICCV 2017 https://arxiv.org/abs/1703.10593

**Experience Assessment:**

I have published one or two papers in this area.

**Review Assessment: Checking Correctness Of Derivations And Theory:**

I assessed the sensibility of the derivations and theory.

**Review Assessment: Checking Correctness Of Experiments:**

I did not assess the experiments.

**Review Assessment: Thoroughness In Paper Reading:**

I read the paper at least twice and used my best judgement in assessing the paper.

---

> ### Author Response · Authors · 2019-11-12
> **Rebuttal on (2) “Connections to current algorithms”**
>
> 2 ) “Connections to current algorithms”
> (2.1) “What is the gain of using deep networks with regard to rather old techniques?”
> Note that our analysis focuses on existing DNNs for segmentation that are state-of-the-art, ie. networks with dilated convolutions and with convolutional LSTMs. The use of “old techniques”, namely ray-intersection and the coloring algorithms, is solely for the purpose of mathematically demonstrating that the state-of-the-art DNN architectures can solve the insideness problem with a network’s size that is realizable in practice. Note that our proof is a proof of existence and we do not claim that the solutions we found are unique, ie. it is possible that there are even smaller networks that solve the insideness problem.
>
> (2.2) “connections to recent algorithms of automatic fill-in of images of contours based on conditional GANs”
> We agree with R#1 that the paper [2.2] is related to our insideness work because it is a potential application of insideness in natural images. Also, the paper [2.2] helps motivating our work, as it is unclear if the DNN in [2.2] (which is a DNN for segmentation, FCN) uses insideness and captures the long-range dependencies in the image, or solely exploits biases in the training set that do not generalize in novel images. We have cited [2.2] in the Introduction. Thank you for pointing us to this interesting work.

---

> ### Author Response · Authors · 2019-11-12
> **Rebuttal on (1) “Insideness in discontinuous curves”:**
>
> (1) “Insideness in discontinuous curves”:
> The Gestalt law of closure shows that human subjects tend to perceive shapes as being whole even when parts of the shapes are missing, as human perception fills in the visual gap. Our definition of insideness does not take into account the Gestalt’s law of closure because in our definition, if there is a discontinuity in the curve all the image would be considered as “outside” region, ie. the “inside” region requires a complete closure of the curve. This simplification is because of the reductionist approach we have used in the paper, which isolates insideness from other factors and facilitates its analysis. Now that we have gained some understanding of the generalization capabilities of existing DNNs for insideness, we are ready to explore a more sophisticated version of insideness in future works. The Gestalt’s law of closure is a very interesting research direction. We have added this in the paper in section 2 and future work, jointly with the other factors we already commented (eg. the representation of the hierarchy of segments).

---

> ### Author Response · Authors · 2019-11-12
> **Rebuttal**
>
> We thank the reviewer for this very valuable and insightful review. In the following, we answer the reviewer’s questions.

---

### Official Review · AnonReviewer5 · 2019-11-04
**Official Blind Review #5**

**Rating:** 3

**Review:**

This paper investigates the problem of modeling insideness using neural networks. To this end, the authors carefully designed both feedforward and recurrent neural networks, which are, in principle, able to learn the insideness in its global optima. For evaluation, these methods are trained to predict the insideness in synthetically generated Jordan curves and tested under various settings such as generalization to the different configuration of curves or even different types of curves. The experiment results showed that the tested models are able to learn insideness, but it is not generalizable due to the severe overfitting. Authors also demonstrated that injecting step-wise supervision in coloring routine in recurrent networks can help the model to learn generalizable insideness.

This paper presents an interesting problem of learning to predict insideness using neural networks, and experiments are well-executed. However, I believe that the paper requires more justifications and analyses to convince some claims and observations presented in the paper. More detailed comments are described below.

1. Regarding the usefulness of learning insideness to improve segmentation
The authors motivated the importance of learning insideness in terms of improving segmentation (e.g., instance-wise segmentation). However, I believe that this claim is highly arguable and needs clear evidence to be convincing. Although I appreciate the experiments in the supplementary file showing that some off-the-shelf segmentation models fail to predict insideness, I believe that these two are very different tasks (one is filling the region inside the closed curve and the other is predicting the labels given the object texture and prior knowledge on shapes; please also note that segmentation masks also can be in very complex shapes, where the prior on insideness may not be helpful). It is still weak to support the claim that learning to predict insideness is useful to improve segmentation.

2. More analyses for experiment results
In the experiment, the authors concluded that both feedforward and recurrent neural networks are not generalized to predict insideness in fairly different types of curves. However, it is hard to find further insights in the experiments, such as what makes it hard to generalize this fairly simple task. Improving generalization using step-wise supervision in a recurrent neural network is interesting but not surprising since we simply force it to learn the procedure of predicting insideness using additional supervision.

To summarize, although the problem and some experiment results presented in the paper are interesting, I feel that the paper lacks justifications on the importance of the problem and insights/discussions of the results.


**Experience Assessment:**

I have published one or two papers in this area.

**Review Assessment: Checking Correctness Of Derivations And Theory:**

I assessed the sensibility of the derivations and theory.

**Review Assessment: Checking Correctness Of Experiments:**

I assessed the sensibility of the experiments.

**Review Assessment: Thoroughness In Paper Reading:**

I made a quick assessment of this paper.

---

> ### Author Response · Authors · 2019-11-12
> **Rebuttal on 2. “More analyses for experiment results”**
>
> 2. “More analyses for experiment results”
>
> Note that the paper provides insights about why DNNs do not generalize in the subsection of 4.2 called "visualization” of the initial submission. We show that the neurons of the feed-forward networks are tuned to the features of the curves in the dataset and there are no signs that they capture the long-range dependencies necessary for solving insideness in general. Also, we found that the recurrent networks expand the inside/outside regions starting from the curve, resulting in only local features being used to determine the direction of expansion. Thus, the DNNs that we evaluated do not generalize because they learned solutions that do not take into account the long-range dependencies in an effective way. These learned solutions are sufficient to achieve high accuracy in the family of curves seen during training, but they do not  generalize to other curves. Then, in section 4.3 we show that the learning strategy can be constrained with stepwise learning in order to encourage that the learned solution captures the long-range dependencies and can generalize. We have reworded these sections to make clear the insights we provide.
>
> Regarding that R#5 does not find surprising that the stepwise learning improves the generalization capabilities, we would like to emphasize the massive gains of accuracy yielded by this strategy. Observe that the stepwise training leads to a cross-dataset accuracy of almost 100% while with the standard learning the cross-dataset accuracy is only ~20% in the best case. In the revised version of the paper, we have emphasized this massive improvement by splitting Fig.5b into two: one for the cross-dataset evaluation and the other for the within dataset evaluation (moved to the appendix). It can now be seen after a quick assessment that the improvement of the generalization capabilities with the step-wise learning is very large. We believe this is a non trivial observation, given that stepwise learning has not been used in any of the state-of-the-art learning strategies.

---

> ### Author Response · Authors · 2019-11-12
> **Rebuttal on 1.  “Usefulness of learning insideness to improve segmentation”**
>
> 1.  “Usefulness of learning insideness to improve segmentation”
>
> We agree with R#5 that segmentation in natural images may involve different cues than insideness. This was commented in the introduction: "[in semantic segmentation benchmarks],  insideness  is  not  necessary  since a solution can rely only on object recognition." Also, we agree with R#5 that segmentation is not the same as insideness, eg. in the introduction we mention: “[In this paper,] we take the reductionist approach by isolating insideness from other components in image segmentation.”
>
> Yet, note that the motivation of analysing insideness is to understand the generalization capabilities of existing segmentation architectures beyond current benchmarks in natural images. This motivation arises from the recent trend of tackling more sophisticated segmentation tasks, eg. segmentation in images that lack texture or color (as in cartoons and sketches) or with unfamiliar objects (such as objects with different textures from those seen during training), in new tasks that require more sophisticated visual spatial reasoning (such as containment or instance-aware segmentation), etc. Note that insideness is a key component for image segmentation in such general settings. Analysing insideness in isolation is a step towards solving these more challenging segmentation problems. Thus, the motivation of this work goes beyond improving DNNs in the current benchmarks (although improvements in these benchmarks with insideness can not be discarded, as pointed out in “future work”). We have reworded the Introduction in order to further clarify these points.
>
> We think that R#5’s concern can be also resolved with R#2's comments, who has “read the paper thoroughly” (quoting R#2): "This work is not like other segmentation publications that just propose a network and start training, but perform some deep analysis about the generalization capability of the existing network architectures.", "It helps other researchers to rethink the boundary problem by using the insideness concept. I think this work will have an impact in semantic segmentation field." and “This paper is well written and well motivated.”

---

> ### Author Response · Authors · 2019-11-12
> **Rebuttal**
>
> We thank R#5 for all the comments and for pointing us what she/he finds unconvincing. This review has been valuable for improving the paper and in the following we address R#5’s concerns.

---

### Author Response · Authors · 2020-04-08
**Paper published at Neural Computation**

https://doi.org/10.1162/neco_a_01413

*The paper in the ICLR website is not updated*

---

### Decision · Program_Chairs · 2019-12-19

**Decision:**

Reject

**Comment:**

This paper investigates a notion of recognizing insideness (i.e., whether a pixel is inside a closed curve/shape in the image) with deep networks. It's an interesting problem, and the authors provide analysis on the limitations of existing architectures (e.g., feedforward and recurrent networks) and present a trick to handle the long-range relationships. While the topic is interesting, the constructed datasets are quite artificial and it's unclear how this study can lead to practically useful results (e.g., improvement in semantic segmentation, etc.).